# Electrochemical Degradation of Diuron by Anodic Oxidation on a Commercial Ru$_{0.3}$Ti$_{0.7}$O$_2$ Anode in a Sulfate Medium

Lucas B. de Faria [1], Guilhermina F. Teixeira [1], Andréia C. F. Alves [2], José J. Linares [3], Sérgio B. Oliveira [2], Artur J. Motheo [4] and Flavio Colmati [1,*]

1   Instituto de Química, Universidade Federal de Goiás, Campus Samambaia, Goiânia 74001-970, Brazil; mina.guilher@gmail.com (G.F.T.)
2   Instituto Federal de Goiás, Campus Goiânia, Goiânia 74055-110, Brazil; dr_botelho@yahoo.com.br (S.B.O.)
3   Instituto de Química, Universidade de Brasília, Campus Universitário Darcy Ribeiro, Brasília 70910-900, Brazil; joselinares@unb.br
4   Instituto de Química de São Carlos, Universidade de São Paulo, São Carlos 13566-590, Brazil; artur@iqsc.usp.br
*   Correspondence: colmati@ufg.br

**Abstract:** This work presents the electrochemical degradation of the herbicide Diuron by anodic oxidation on a Ti/Ru$_{0.3}$Ti$_{0.7}$O$_2$ metal mixed oxide anode using sulfate as the electrolyte. The study includes the influence of Diuron concentration and current density on anodic oxidation. The results evidence a first-order degradation, with the highest capacity achieved at 40 mA cm$^{-2}$ and at an initial Diuron concentration of 38 mg L$^{-1}$. Nevertheless, in terms of efficiency and energy demand, the operation at 10 mA cm$^{-2}$ is favored due to the more efficient and less energy-consuming condition. To discern the optimum design and operation conditions, this work presents the results of a preliminary technical–economic analysis, demonstrating that, to minimize the total costs of the system, it is recommended to seek the most efficient conditions, i.e., the conditions demanding the lowest applied charges with the highest Diuron degradation. At the same time, attention must be given to the required cell voltage to not increase excessively the operating costs.

**Keywords:** MMO; Diuron; anodic oxidation; sulfate; operating parameters





## 1. Introduction

Water contamination by manufactured organic chemicals such as herbicides, pesticides, dyes, and antibiotics demands efforts to design appropriate treatment technologies. Classical wastewater treatment plants (WWTP) are based on physical and biochemical processes, which cannot remove persistent pollutants [1]. In this sense, advanced oxidation processes (AOP) emerge as a suitable alternative based on the generation of the powerful oxidant hydroxyl radical (OH•) [2,3]. Among the different AOP, electro-oxidation presents some advantages, such as adding few or no chemicals to the medium, versatility, easy automation, high energy efficiency, and the capacity to mineralize the pollutants completely [4].

In the literature, recent works show the high potential of the AOP to deal with refractory pollutants. For instance, Jian et al. [5] prepared hierarchical dendritic Co$_3$O$_4$-SnO$_2$ nanostructures by hydrothermal treatment with a high electrochemically active surface, low charge transfer resistance, and high photoelectrochemical activity. The researchers successfully applied this material to the degradation of Reactive Brilliant Blue KN-R. Ji et al. [6] prepared Cu nanoparticles supported by N-doped TiO$_2$ by pre-anchor and post-pyrolysis strategy of Cu-MOF structures. The prepared materials were applied to the electrochemical reduction of NO$_3^-$ to NH$_3$, in which the formation of copper nitride combined with Cu is crucial to maximize the performance and the faradaic efficiency. Ma et al. [7] prepared a binary Co$_3$O$_4$-δ-MnO photoelectrocatalyst and applied this to the degradation of the

reactive brilliant blue KNR, showing the importance of the preparation procedure to obtain an active material with the capacity to degrade the dye up to a level of 97% with excellent durability. The authors attributed this outstanding performance to the hollow structures produced during the synthesis, generating a very active surface.

Herbicides are a group of refractory chemicals that deserve attention due to their extensive worldwide application in intensive agriculture. Diuron (3-(3,4-dichlorophenyl)-1,1-dimethyl urea) is a systemic herbicide that effectively controls weeds, broadleaf, and grasses. It belongs to the arylurea class, acting by the inhibition of photosynthesis [8]. Diuron, present in drinking waters, is known to be potentially carcinogenic [9], mandatorily demanding adequate treatment. As Diuron is not readily biodegradable [10], physicochemical removal methods emerge as suitable candidates for the Diuron treatment, particularly advanced oxidation processes. As recently reviewed [11], ultrasound-assisted processes are effective in degrading herbicides/pesticides as an assistant to Fenton, anodic oxidation, and photocatalysis. Metal–organic-framework-based catalysts are also effective in degrading this type of refractory pollutants by sulfate radical generation or coupling to Fenton processes [12]. Natural minerals are also effective catalysts for degrading herbicides and pesticides, especially when surface defects are present in the mineral microstructure [13].

Anodic oxidation is based on applying an electrical current or potential difference to an electrolytic solution (it could be the residue that is treated if it possesses enough conductivity) between an anode, where pollutant degradation occurs, and a cathode, where hydrogen evolution evolves [14]. Two types of anodes are used in these studies. In non-active electrodes, the water discharge reaction produces hydroxyl radicals ($\bullet OH$) physisorbed on the surface of the anode (e.g., boron-doped diamond (BDD) and $PbO_2$). The weak interactions between the electrode surface and the formed radical (Van der Waals forces) allow the formed radical to detach from the surface and easily interact with species in the proximities of the surface. The other type is the active anodes, represented by mixed metallic oxides (MMO), such as $RuO_2/TiO_2$, $IrO_2/TiO_2$, $SnO_2/TiO_2$, and $Sb_2O_5$, in which the hydroxyl radical remains chemisorbed on the surface, favoring the oxygen evolution reaction [15]. In terms of oxidative power, BDD is recognized as the best material, given its capacity to produce large amounts of $OH\bullet$, as well as high chemical resistance, long-time stability, low capacitive current, and a wide electrochemical window [16]. Nevertheless, BDD anodes possess two primary counterparts: (i) a significant initial investment [17]; (ii) the formation of perchlorate in the presence of chloride [18,19]. These BDD shortcomings have stimulated the development and application of MMO to the degradation of refractory pollutants as an alternative anode [20–23].

The application of MMO to herbicides/pesticides degradation has been successfully demonstrated. Zhao et al. [24] studied different types of $Ti/IrO_2$ MMO, combined with $RuO_2$, $SnO_2$-$Ta_2O_5$, and $Ta_2O_5$, evidencing that the most suitable material was that with $Ta_2O_5$, achieving levels of degradation close to 90% of Nicosulfuron under optimum conditions, 0.8 A, pH 3, and interelectrode space of 3 cm, pointing out the importance of selecting adequately the operating conditions. Malpass et al. [25] studied MMO composed of a mixture of $RuO_2$ and $SnO_2$. In their study, the authors evidenced the importance of photo-assistance for degrading the pesticide atrazine and a correlation between the morphology of the electrodes and the chemical oxygen demand degradation. Espinoza et al. [26] prepared nanosized $IrO_2$ and $RuO_2$ MMO separately and in the same reaction medium. They observed that the materials in which each oxide was separately synthesized degraded more effectively the oxamic acid. In a novel approach, Santos et al. [27] prepared MMO electrodes with $RuO_2/IrO_2/SnO_2/CeO_2$ by the Pechini method and by the thermal decomposition of the metallic chlorides dissolved in an ionic liquid. The researchers observed that the material $Ti/(RuO_2)_{0.8}(IrO_2)_{0.2}$ prepared with ionic liquid was the most active composition due to the intrinsic high activity of the materials and the rough surface arising from the ionic liquid method. Malpass et al. [28] used a MMO formed by $Ti/Ru_{0.3}Ti_{0.7}O_2$. They studied the influence of the electrolyte (NaCl, NaOH, $NaNO_3$, $NaClO_4$, $H_2SO_4$, and $Na_2SO_4$). The researcher observed that NaCl outperformed

the other electrolytes and could mineralize atrazine. No advantage was seen if $SnO_2$ replaced $Ru_{0.3}Ti_{0.7}O_2$. Vlyssides et al. [29] degraded four commercial pesticides (Demeton-*S*-methyl, Metamidophos, Fenthion, and Diazinon), observing, in a Ti/Pt anode bench- and pilot-plant scale reactor, that all the pesticides were partially degraded. Nevertheless, a notable increase in biodegradability was observed, so they recommended using the anodic oxidation as a detoxification step. Arapoglou et al. [30] degraded methyl-parathion in the same system as Vlyssides et al. [26], observing a significant decrease in the chemical and biological oxygen demand, especially in acidic conditions. They estimated the energy demand in 8–18 kWh kg$^{-1}$ of COD reduced. Finally, Malpass et al. [31] degraded the pesticide carbaryl by electrochemical and photo-assisted electrochemical methods. Two significant findings were the synergistic effect of UV light in degrading the pesticide and the non-necessity of using NaCl as supporting electrolyte, avoiding the appearance of the organochlorinated.

Electrochemical remediation of Diuron has not been extensively explored. Zheng et al. [32] recently presented the results of some herbicides removal—Diuron included—by anodic oxidation on graphite at low voltages. The authors elucidated that the main oxidant present in the medium was the superoxide radical. Rahmani et al. [33] achieved the complete mineralization of Diuron by treatment with the non-active $PbO_2$ anode and suspended granular activated carbon, acting as bipolar electrodes. Zheng et al. [34] applied a solar-driven electro-oxidation in three different anodes: titanium, graphite, and BDD, observing that the most efficient and stable performance, attaining 90% of Diuron elimination, was obtained by BDD. Pipi et al. [35] analyzed the Diuron degradation on MMO anodes, studying the influence of the composition ($IrO_2$ vs. $RuO_2$). $RuO_2$ emerged as the most active material. Khongthom et al. [36] studied the Diuron degradation on a graphite anode in a microscale reactor with levels of degradation above 90% for residence times of 100 s. Zhu et al. [37] achieved a 100% Diuron degradation on a $Co_3O_4$/graphite composite anode at pH 2.0 and 2 mA cm$^{-2}$. Finally, Bumroongsakulsawat et al. [38] analyzed the influence of sulfate and nitrate interferences on the Diuron degradation mechanism and toxicity of the final effluent. The researchers observed that both anions reduced the efficacy of Diuron oxidation by scavenging hydroxyl radicals. It is important to note that most of these studies used chloride salts as supporting electrolytes, which may give rise to the formation of organochlorides [39], an issue that deserves attention.

With these antecedents, this manuscript addresses the Diuron degradation on a commercial electrochemical reactor equipped with a $Ru_{0.3}Ti_{0.7}O_2$ commercial MMO anode (De Nora Corp., Sorocaba, Brazil) using sodium sulfate as the supporting electrolyte. Differently from most of the previous studies in which NaCl is used as supporting electrolyte, we propose substituting it with $Na_2SO_4$ to verify the activity of the commercial MMO towards Diuron oxidation. In this way, we expect that the appearance of intermediate organochlorines might be avoided, verifying the oxidizing capacity of this anode. This work comprises the study of some critical operating parameters, such as the current density and the initial Diuron concentration. The kinetics of the degradation under the different operating conditions and the energy demand are analyzed to establish the optimum operating conditions in terms of a large Diuron removal combined with the lowest possible energy consumption. Finally, a preliminary technical–economic assessment is presented to select the most suitable conditions for designing and operating this system. This study is mandatory to minimize the total costs of the electrochemical reactors, considering the annualized capital costs (from the electrodes) and the operating costs (electricity consumption).

## 2. Materials and Methods

Sodium sulfate and Diuron were purchased from Sigma-Aldrich (Sigma-Aldrich Brazil Ltd., São Paulo, Brazil) and used as received. Ultrapure water was obtained from a Millipore Milli-Q system (resistivity = 18 MΩ cm, total organic carbon below 2 μg L$^{-1}$). The synthetic wastewater polluted with Diuron was prepared by initially dissolving $Na_2SO_4$ in the ultrapure water to render a 0.17 mol L$^{-1}$ $Na_2SO_4$ solution (ionic strength = 0.5).

Afterward, Diuron was dissolved in the different studied concentrations, 9.5, 19, and 38 mg L$^{-1}$. Higher concentrations were unfeasible due to the low solubility of Diuron in water [40] (42 mg L$^{-1}$).

The electrochemical reactor consisted of two electrodes, an MMO-Cl anode (Ru$_{0.3}$Ti$_{0.7}$O$_2$ supported on Ti) and a stainless-steel cathode with a geometric area of 15 cm$^2$. The interelectrode gap was 2 cm, sealing the reactor with a silicone gasket. A peristaltic pump (IntLLab$^{TM}$) was used to apply a flow rate of 4 L h$^{-1}$ operating in recirculation mode. The electrolysis experiments were carried out by connecting the reactor to a power supply EMG 18135 30 A (Orion-EMG Elektronikus, Budapest, Hungary). The current was measured by a Minipa DT-930B multimeter (Minipa do Brasil Ltd., São Paulo, Brazil), and the cell voltage was measured by a Minipa ET-1002 multimeter (Minipa do Brasil Ltd., São Paulo, Brazil).

The electrolysis studies were also performed at different current densities, 10, 20, and 40 mA cm$^{-2}$, collecting aliquots at different degradation times for UV-VIS characterization. The sequence and label of the applied conditions are summarized in Table 1.

**Table 1.** Experiments carried out in this study and their corresponding labels.

| | | Diuron Concentration (mg L$^{-1}$) | | |
| --- | --- | --- | --- | --- |
| | | **9.5** | **19** | **38** |
| Current density (mA cm$^{-2}$) | 10 | D10-9.5 | D10-19 | D10-38 |
| | 20 | D20-9.5 | D20-19 | D20-38 |
| | 40 | D40-9.5 | D40-19 | D40-38 |

Diuron was quantified by UV-VIS spectroscopy. The herbicide presents an absorption peak at 250 nm, which allows for the monitoring of Diuron degradation, which decreased during the experiments. Finally, the parameters applied the volumetric charge (Q) and the energy demand (E), estimated from Equations (1) and (2), respectively, where E is the volumetric energy consumption, V$_{cell}$ is the cell voltage, I is the applied current, t is the degradation time, and $\upsilon$ is the volume of the electrochemical reactor:

$$Q\left(\text{Ah L}^{-1}\right) = \frac{\text{I(A)t(h)}}{\upsilon(\text{L})} \tag{1}$$

$$E\left(\text{kWh m}^{-3}\right) = \frac{0.001\ \text{V}_{cell}(\text{V})\ \text{I(A)t(h)}}{\upsilon(\text{m}^3)} \tag{2}$$

The Diuron degradation experimental data were fitted to a first-order degradation kinetics according to Equation (3), where C is the Diuron concentration at any time (t), k is the rate constant, and C$_0$ is the initial Diuron concentration. Its integration gives rise to Equation (4), which defines the exponential decay of the Diuron concentration:

$$\frac{\text{dC}}{\text{dt}} = -\text{kC} \tag{3}$$

$$C = C_0 \exp(-\text{kt}) \tag{4}$$

## 3. Results and Discussion

Figure 1 shows the Diuron removal for the different degradation times at the studied current densities and initial Diuron concentrations. As can be seen, the maximum Diuron removal is achieved for the operation at the highest current density and initial Diuron concentration, D40-38. These two combined conditions render the most oxidative degradation environment with the highest availability of the target molecule, resulting in the largest degraded Diuron at any time. The operation at lower current densities and/or smaller initial Diuron concentrations results in a more sluggish Diuron degradation. This behavior is typical of mass-transfer-controlled processes, where the more limited access

of the Diuron molecule to the electrode surface or the available electrogenerated oxidants (OH• radicals, peroxide, persulfate given the supporting electrolyte used, ozone, etc.) favors the development of ineffective parasitic reactions (e.g., oxygen evolution). In fact, the operation at low current densities reduces the extension of parasitic secondary reactions, increasing the efficiency of Diuron oxidation for the lowest Diuron concentration condition (experiment D10-9.5) [41]. In general terms, the operation at higher Diuron initial concentrations for a fixed current density increases the Diuron removed due to a more efficient process. The effect of the initial Diuron concentration is more dependent on the applied current density. A higher initial concentration (DXX-38) allows for operating effectively at higher current densities, whereas operating at DXX-9.5 degrades Diuron more rapidly at lower current densities, as discussed above. It is also interesting to analyze the results regarding the percentage of Diuron removed. The corresponding results are collected in the Supplementary Material (SM, Figure S1). Regarding the relative Diuron degradation, the operation in the condition D40-19 renders the highest percentage.

Figure 2 shows the evolution of the degraded Diuron with the applied volumetric charge (Equation (2), where Q is the applied volumetric charge). This figure allows us to visualize the efficacy of Diuron degradation since it compares the applied charge with the actual degradation of Diuron. As can be observed, the most efficient degradation is reached for the D10-38 experiment, operated at the lowest current density and higher initial Diuron concentration. This condition disfavors the evolution of ineffective reactions, which, combined with the high target pollutant concentration, results in more efficient degradation. On the contrary, the condition D40-9.5 results in the least effective degradation due to the smallest target pollutant availability combined with the operation at high current density. Higher initial concentrations and lower current densities increase the efficiency of Diuron degradation. Nevertheless, caution should be taken as this would lead to longer operation times and higher electrode areas, demanding a detailed and systematic techno–economic analysis.

Figure 3 displays the rate constant of the degradation process. The concentration profiles are shown in the SM (Figure S2). The results fit well with a first-order kinetic, which is expected for mass-transfer-controlled degradation [42]. The rate constant values reveal that the fastest degradation occurs for the D40-19, as observed in Figure S1, with the highest use of the available oxidants. In contrast, the D40-9.5 results in the slowest degradation due to the significant evolution of parasitic reactions. Not as fast as operating at the optimum, operating at D10-9.5, D20-19, and D40-38 also render a fast Diuron degradation kinetic. In these conditions, the oxidants are also effectively utilized.

Another relevant parameter of anodic oxidation is the energy consumption. Figure 4 depicts the energy consumption as a function of time and Diuron removed. In terms of energy demand, as expected, the operation at low current densities is favorable due to the smallest values of the cell voltage accompanied by the highest degree of Diuron removal at D10-38. Under this condition of high target pollutant availability and low current density, we can oxidize more effectively the Diuron molecule with a less energy-costly process. Note that for all experiments, the final concentrations of Diuron were close to 67% of the initial concentration (Figure S2); thus, the energy consumption can be assigned totally to Diuron degradation.

In summary, regarding Diuron degradation, these results demonstrate that Diuron can be anodically degraded in MMO. Compared with the studies mentioned in the Introduction section [32–38], we can infer that our results are inferior in terms of removal efficiency (we achieve removals up to 40% compared with 90% reported in the mentioned studies). Nevertheless, we need to consider that this study proposes using a sulfate medium instead of chloride to assess the feasibility of degrading Diuron, intending to avoid the generation of organochlorines (current ongoing research). Furthermore, we are exploring other aspects, such as the identification, quantification, and toxicity assessment of the formed products, to gain a more complete knowledge of the MMO performance on degrading Diuron.

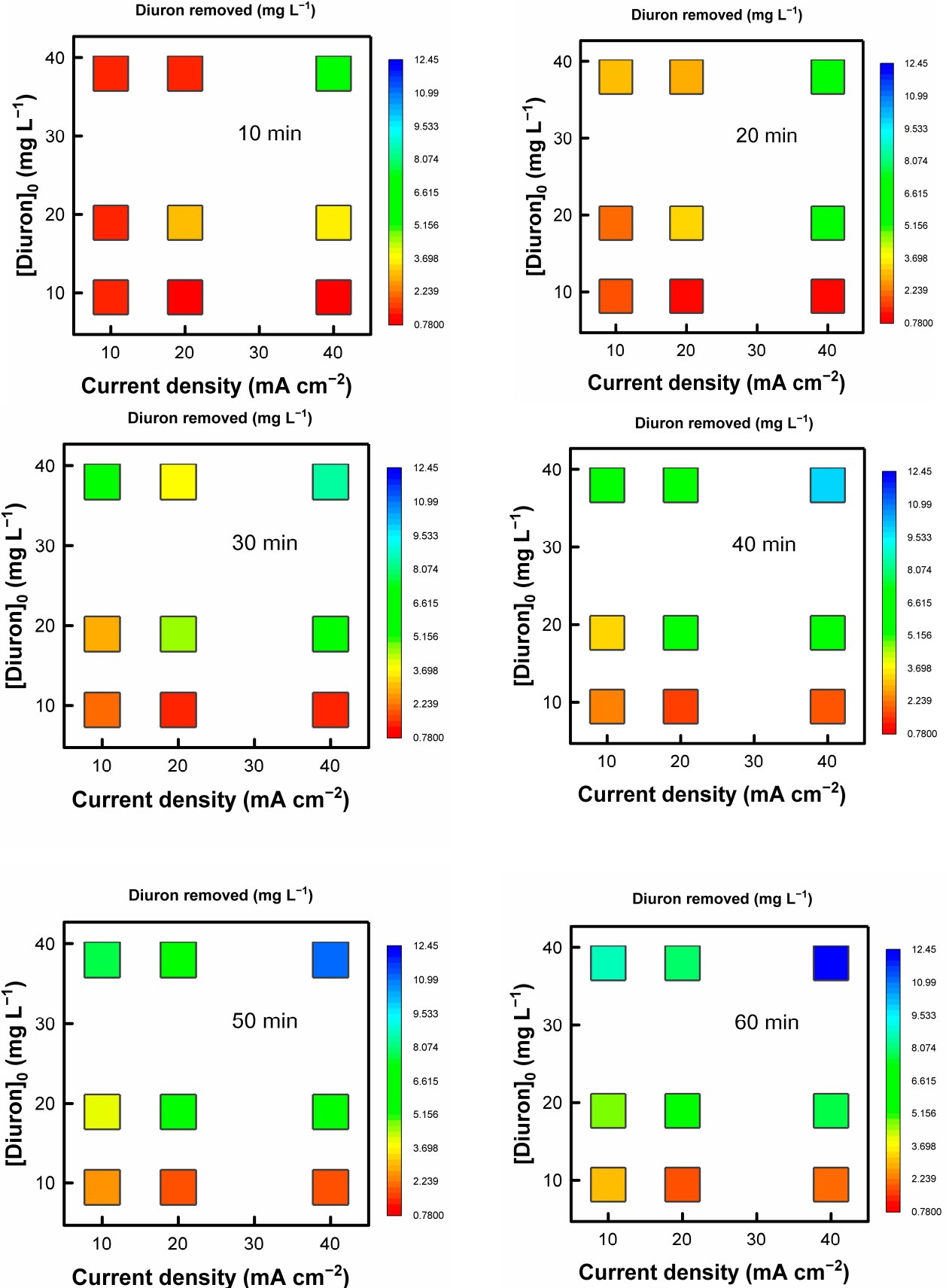

**Figure 1.** Evolution of the removed Diuron for the experiments carried out at different initial Diuron concentrations and current densities.

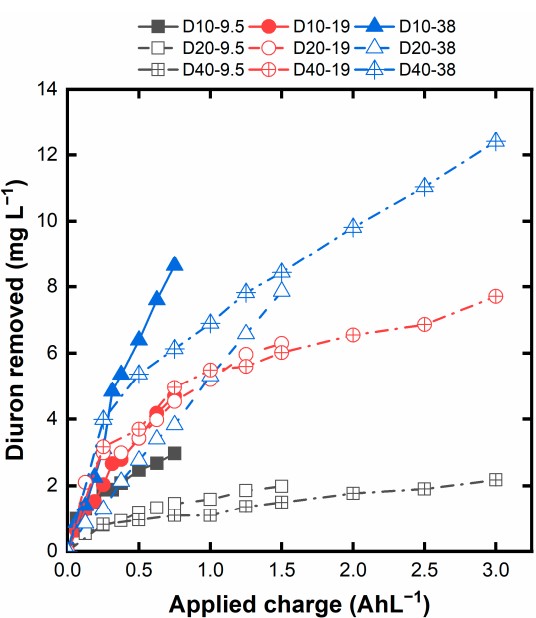

**Figure 2.** Diuron removal for the different applied charges in the experiments.

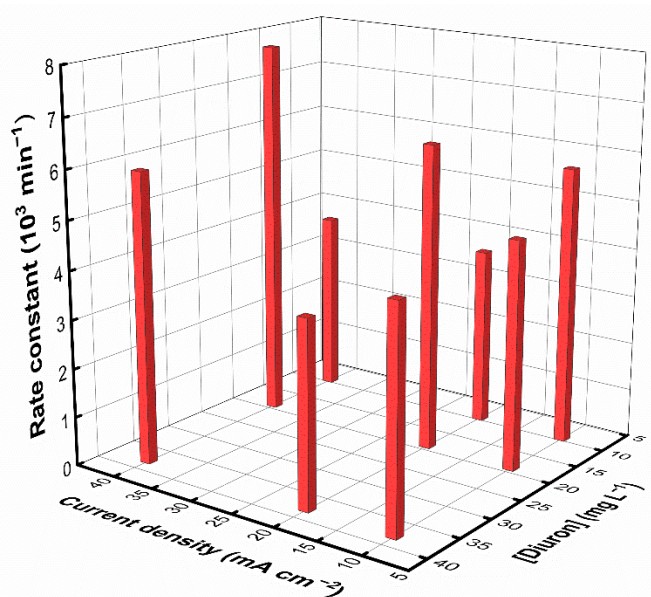

**Figure 3.** Kinetics rate constant for the degradation process.

Caution must be taken when analyzing these results since, during treatment at high initial concentrations, the Diuron concentration drop is expected to make the process less efficient, increasing the energy demand as the degradation proceeds. Such an effect could, in the end, make the process more onerous. In this sense, we present a preliminary and straightforward estimate of the system cost based on the required area and the energy consumption, given that these two parameters account for most of the system's fixed capital and operation costs [43]. The MMO cost can be estimated from the value provided by Wenderich et al. [17], estimated at US$ 3500.00 per m$^2$ of electrode. The cost of electricity ($C_{elec}$) is considered to be 0.0845 US$ kWh$^{-1}$ [44]. To design the system, it is necessary to know the required charge to degrade the target molecule. Let us consider a treatment plant whose capacity is equal to that presented by Cabral Coelho and Santos Brega [45], 1 m$^3$ h$^{-1}$, with an effluent containing 38 mg L$^{-1}$ of Diuron. Based on the results of Kučić Grgić et al. [46] regarding the influence of the initial Diuron concentration in an effluent on its biodegradation, we can consider a treatment down to 9.5 mg L$^{-1}$ (Diuron could be

biologically degraded below 10 mg L$^{-1}$). Thus, the design of the treatment plant will be executed based on the two initial concentrations in successive sequences, i.e., from 38 down to 19 mg L$^{-1}$ (Stage 1), and 19 down to 9.5 mg L$^{-1}$ (Stage 2), after which we will achieve the goal for the biological treatment. The required electrode area can be estimated from Equation (5), where j is the applied current density. In this case, the parameter υ is the volume to be treated. If we consider three operating cycles, including 1 h for the charge and 1 h for the discharge of the storage tank that contains the solution to be treated, each cycle would have an operating time of 6 h to treat 8 m$^3$. Finally, the parameter j is the applied current density.

$$A\left(m^2\right) = \frac{Q\left(A\,h\,L^{-1}\right)\upsilon(L)}{j\left(A\,m^{-2}\right)t\,(h)} \tag{5}$$

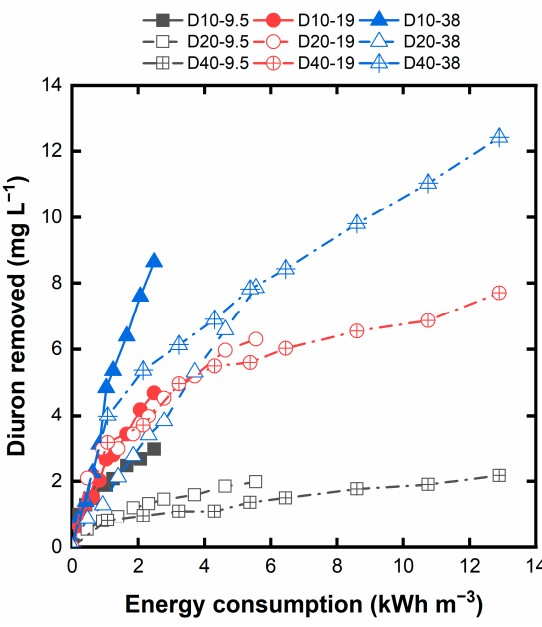

**Figure 4.** Energy consumption for the different degradation processes.

To establish the required charge for Stage 1 and Stage 2, we have tentatively fitted the experimental data to an empirical power equation ($y = a\,x^b$), whose fitting is shown in the SI (Figure S3). Furthermore, Table S1 collects the values of the required volumetric charges. Table 2 presents the required MMO electrode areas and their corresponding fixed capital costs (FCC) based on this latter parameter, the operation time, and the current density. In order to compare the FCC with the operating costs (OC), the FCC are annualized by dividing by three, according to Towler and Sinnott [47].

**Table 2.** Required electrode area of the designed plant, corresponding total, and annualized costs (M\$ = 10$^6$ \$).

|  |  | A (m$^2$) | FCC (M\$) = 10$^{-6}$ × A × 3500 \$ m$^{-2}$ | Annualized FCC (M\$ year$^{-1}$) = 0.33 × CC |
|---|---|---|---|---|
| Current density (mA cm$^{-2}$) | 10 | 146.6 | 0.513 | 0.164 |
|  | 20 | 140.4 | 0.492 | 0.157 |
|  | 40 | 128.4 | 0.449 | 0.144 |

As can be observed, the increase in the applied current density reduces the required electrode area as the required charges do not increase at the same rate as the current density. This is beneficial from the point of view of the initial investment of the plant. Nevertheless,

attention must be given to the operational costs, exclusively estimated from the electricity consumption. Equation (6) allows us to estimate the operating costs, where OF is the operating factor, considered to be 8000 h year$^{-1}$ corrected by the 6/8 factor corresponding to the 2 h devoted to charge/discharge stages. The cell voltage for the operation at 10 mA cm$^{-2}$ is 3.3 V, at 20 mA cm$^{-2}$ is 3.7 V, and at 40 mA cm$^{-2}$ is 4.3 V.

$$OC\left(\$\ year^{-1}\right) = 0.001 V_{cell}(V) j\left(A\ m^{-2}\right) A\left(m^2\right) OF\left(h\ year^{-1}\right) C_{elec}\left(\$\ kWh^{-1}\right) \quad (6)$$

Table 3 collects the OC for the different current densities, whereas Figure 5 displays the corresponding FCC and OC and the total annualized costs. As observed, the decay in the annualized CC as the current density rises is counterbalanced by the higher OC from the energy consumption. Thus, the optimum condition based on the system's total costs corresponds to the current density of 10 mA cm$^{-2}$.

**Table 3.** OC for the different applied current densities.

|  |  | $V_{cell}$ (V) | j (A m$^{-2}$) | A (m$^2$) | OF (h Year$^{-1}$) | OC (M\$ Year$^{-1}$) |
|---|---|---|---|---|---|---|
| Current density (mA cm$^{-2}$) | 10 | 3.3 | 100 | 146.6 |  | 0.0244 |
|  | 20 | 3.7 | 200 | 140.4 | 6000 | 0.0525 |
|  | 40 | 4.3 | 400 | 128.4 |  | 0.112 |

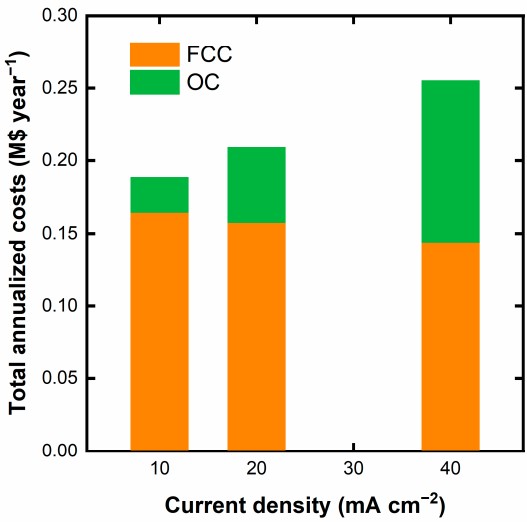

**Figure 5.** Estimation of the CC and OC for the different current densities. The height of each bar represents the total annualized cost.

One interesting feature of the applied charge versus Diuron removed is that at low Diuron removal, the operation at 40 mA cm$^{-2}$ is more efficient. Figure S4 shows in detail the mentioned regions from which we can infer the volumetric charges that could be applied at a current density of 40 mA cm$^{-2}$, whereas the remaining volumetric charge would be applied at 10 mA cm$^{-2}$ for each stage. To minimize the total annualized costs, we have optimized the time at which each current density is applied with the aid of the *Solver* tool of Microsoft Excel. Table 4 summarizes the main results. It is important to note that the electrode area would have a unique value calculated from the highest area obtained in the optimization of the four steps. In contrast, the OC would be different, as the required electrode area could be different, as well as the power consumption. Figure 6 graphically compares the total annualized cost for the more economical option of Figure 5 (10 mA cm$^{-2}$) and the combined current densities to visualize better the positive effect of the operation at different current densities.

**Table 4.** Optimized operation time for each cycle, required electrode area, annualized CC, OC, and total costs (addition of the annualized CC and the OC of the four steps).

| | j (mA cm$^{-2}$) | Q (ALh$^{-1}$) | t (h) | Electrode Area (m$^2$) | Annualized CC (M\$ Year$^{-1}$) | Operative Electrode Area (m$^2$) | OC (M\$ Year$^{-1}$) | Total Annualized Cost (M\$ Year$^{-1}$) |
|---|---|---|---|---|---|---|---|---|
| Stage 1 (38→19 mg L$^{-1}$) | Step 1→40 | 0.250 | 0.148 | | | 102.0 * | 0.00218 | |
| | Step 2→10 | 1.524 | 3.12 | | | 117.4 | 0.0102 | |
| | | | | 117.4 | 0.131 | | | (Total OC = 0.0164) |
| Stage 2 (19→9.5 mg L$^{-1}$) | Step 1→40 | 0.735 | 0.376 | | | 117.4 | 0.00639 | 0.158 |
| | Step 2→10 | 1.15 | 2.36 | | | 117.4 | 0.00769 | |

\* In this case, not all the cells would be operative.

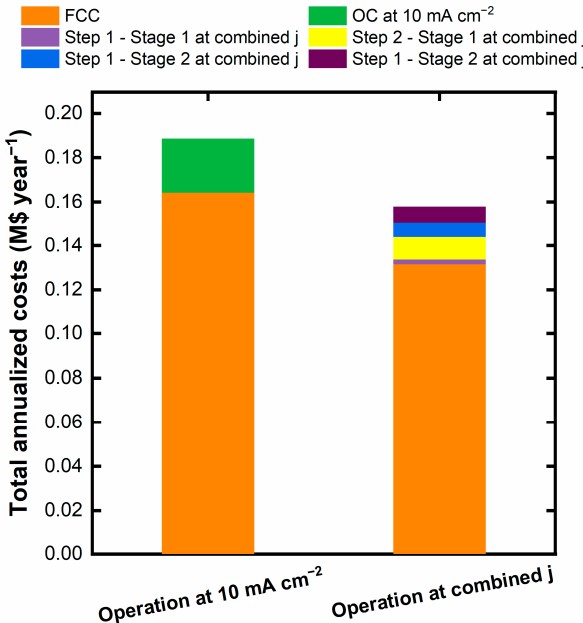

**Figure 6.** Comparison of the CC and OC for the operation at 10 mA cm$^{-2}$ and for the operation with combined current densities.

The results in Figure 6 demonstrate the beneficial effect of operating at combined current densities. The main impact lies in the reduction of the annualized CC due to the reduction in the time operating at the lowest current density, which results in a drop in the required electrode. This area still attends to the required one during the operation at 40 mA cm$^{-2}$. In contrast, the OC slightly increases due to the operation for a short period at 40 mA cm$^{-2}$, not to the point of counterbalancing the drop in the CC.

The results show the necessity of cautiously analyzing the influence of the operating parameters in an electrochemical oxidation system. In addition to the technical–scientific studies, an economic assessment is very advisable to complement the results, discovering the optimum designing and operative conditions to minimize the costs of applying this technology. Finally, for future studies, it could be interesting to extend the treatment times to explore the required time and volumetric charges to achieve higher levels of Diuron degradation. Furthermore, attention must be given to the formed intermediates, their toxicity, identification, and quantification, as well as the oxidant species formed. These more profound studies would provide us with a better perspective of the degradation capacity, toxicity of the formed products, and degradation mechanism. With this more comprehensive knowledge, it will be possible to perform a more accurate economic analysis with a complete study of the investments and the operating costs. Ongoing investigations of the group are exploring all these aspects.

## 4. Conclusions

As a general conclusion, this study has demonstrated that Diuron, an extensively used herbicide, can be removed by anodic oxidation on a $Ti/Ru_{0.3}Ti_{0.7}O_2$ MMO in sodium sulfate. In addition to that, other important conclusions drawn from this study are as follows:

- In terms of a rapid Diuron degradation, the condition D40-38 is the most favorable, owing to the synergy of the most oxidative conditions with the highest pollutant availability, accelerating the degradation process.
- Nevertheless, attention must be given to efficiency and energy consumption, in which case the operation at 10 mA cm$^{-2}$, regardless of the initial Diuron concentration, becomes more attractive as it removes Diuron with the lowest applied charge and energy demand.
- This information drawn from the analysis of the influence of the operating parameters is precious for establishing the design and operating conditions of an electrochemical treatment plant based on the minimization of the costs.
- Operating in the most efficient conditions is beneficial for reducing the FCC, whose impact is more substantial than the OC, provided there is no significant increase in the cell voltage.
- In this latter sense, as demonstrated in the work, it may be interesting to modulate the applied current density during the treatment, beginning with a higher current density and reducing it as the degradation proceeds to guarantee the operation at the highest possible efficiency and lowest electricity consumption, minimizing the total costs of the systems.

**Supplementary Materials:** The following supporting information can be downloaded at: https://www.mdpi.com/article/10.3390/chemengineering7040073/s1, Figure S1: Final percentages of Diuron removed for the different initial Diuron concentrations and current densities studied; Figure S2: Concentration profiles for the different current densities, along with the fitting parameter to a first-order kinetics (C = C$_0$ exp (−kt)); Figure S3: Fitting of the applied charges versus the Diuron removed to estimate the required volumetric charges for designing the treatment system; Figure S4: Fitting of the applied charges versus the Diuron removed to estimate the required volumetric charges for designing the treatment system; Table S1: Required volumetric charges (AhL$^{-1}$) for each stage of Diuron removal for each applied current density.

**Author Contributions:** Conceptualization, L.B.d.F. and F.C.; methodology, L.B.d.F., F.C. and A.J.M.; formal analysis, G.F.T. and A.C.F.A.; investigation, L.B.d.F., G.F.T. and A.C.F.A.; resources, F.C. and A.J.M.; data curation, G.F.T., A.J.M. and J.J.L.; writing—original draft preparation, J.J.L. and S.B.O.; writing—review and editing, F.C. and A.J.M.; visualization, S.B.O. and A.J.M.; supervision, F.C.; project administration, F.C. All authors have read and agreed to the published version of the manuscript.

**Funding:** The authors thank Conselho Nacional de Desenvolvimento Científico e Tecnológico (CNPq), and Coordenação de Aperfeiçoamento de Pessoal de Nível Superior (CAPES), Fundação de Amparo à Pesquisa do Estado de Goiás (FAPEG) and Fundação de Amparo à Pesquisa do Estado de São Paulo (FAPESP) for financial support. LBF thanks CAPES also for the scholarship #88882.386519/2019-01.

**Data Availability Statement:** The data presented in this study are available on request from the corresponding author. The data are not publicly available due to privacy.

**Conflicts of Interest:** The authors declare no conflict of interest.

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
