# Peer review of "Electrochemical Degradation of Diuron by Anodic Oxidation on a Commercial Ru0.3Ti0.7O2 Anode in a Sulfate Medium"

_2305-7084, doi:10.3390/chemengineering7040073_

Round 1
Reviewer 1 Report
In the title “Electrochemical Degradation of Diuron by anodic oxidation on DSA in sulfate medium”, the full name of DSA should be given.
Please check and correct the sentence “Diuron ((3-(3,4-dichlorophenyl)-1,1-dimethyl urea)) is a systemic herbicide used to control the growth of one herbicide used to manage the growth of large and narrow-leaf weed plants”.
Please check and correct the sentence “Two types of anodes are used, the non-active ones, in which the water discharge reaction produces hydroxyl radicals weakly adsorbed onto the anode surface, allowing an intense interaction between the pollutant and the powerful oxidant OH•, allowing a deep pollutant oxidation or even mineralization (boron-doped diamond (BDD) and PbO2); and the active anodes, represented by the mixed metal oxides (MMO), such as RuO2/TiO2, IrO2/TiO2, SnO2/TiO2, Sb2O5, among the most representative semiconductors, platinum and graphite”.
As to the description of studies on application of DSA to the degradation of herbicides/pesticides in Introduction, the authors simply list several related studies. It is suggested to summary the studies, discussing the research advances and current challenge.
The innovation of this work should be clearly described in Introduction. Compared to previous studies, why this work is imperative.
References are suggested to support the introduce backgrounds about advanced oxidation processes: Bibliometric analysis and literature review of ultrasound-assisted degradation of organic pollutants. Science of The Total Environment, (2023).162551; Metal-organic frameworks-derived catalysts for contaminant degradation in persulfate-based advanced oxidation processes. Journal of Cleaner Production, (2022). 134118; Harnessing the power of natural minerals: A comprehensive review of their application as heterogeneous catalysts in advanced oxidation processes for organic pollutant degradation. Chemosphere, (2023). 139404.
In the sentence “Figure 1 shows the Diuron removed for the different degradation times at the studied current densities and initial Diuron concentrations”, “removed” should be corrected as “removal”.
The results shown in figure 1 are duplicate with figure 2. Figure 1 should be deleted. As to the results, quantitative description should be added in the manuscript.
The linear fitting results of first-order kinetics should be added, and linear correlation coefficient should be provided to verify the first-order kinetic model.
The kinetic model, calculation of applied volumetric charge, fixed capital costs, and operating costs should be described in “2. Materials and Methods”.
Comparison of the results with previous reports should be listed in a table with removal efficiency and process cost.
The characterizations of the used anode should be added, as well as the studies on mechanism of electrochemical degradation of Diuron by anodic oxidation in the system. Or this manuscript should be refined to be short paper such as letter or communication rather than full article.
In-depth discussion of the results should be added to improve the manuscript.
Grammar errors should be checked and corrected in the whole manuscript.
Grammar errors should be checked and corrected in the whole manuscript.
Reviewer 2 Report
How did you choose Diuron concentrations for this work? Did you use concentrations that has been used with other anode materials? For example when BDD was used as anode?
I could not find file with Supplementary Material!
I suposse that something is wrong/ in contradiction in this sentence, page 5, last paragraph:
As can be observed, the most efficient degradation is reached for the D10-9.5 experiment, operated at the lowest current density and higher initial? Diuron concentration, conditions that disfavor the evolution of ineffective reaction, along with the high concentration of the target pollutant.
Discussion regarding Figure 3 should be about rate constant not about constant rate!!! In Figure that should be changed as well, and in the remaining text.
In Table 3 for current density 40mA/cm2 OF(h/year) is mistake? 0.449 h/year?
You made interesting calculations of Annual costs for Diuron degradation using just one anode material. Can you make comparison with other anode materials? Is this electrochemical process exceptional in some way, chosen to be the most payable? Or with some other Diuron degradation way.
Best regards
It can be improved
Reviewer 3 Report
This work presents a first attempt at the electrochemical degradation of the herbicide Diuran using a sulfate electrode. The work sounds scientifically relevant and is well organized. A careful review of the English language is required.
The work sounds scientifically relevant and is well organized. A careful review of the English language is required.
Reviewer 4 Report
Flavio Colmati and co-workers have a study that provides important insights into the electrochemical degradation of Diuron using a Ti/Ru0.3Ti0.7O2 dimensionally stable anode (DSA) and sulfate as the electrolyte. The investigation of the influence of Diuron concentration and current density on anodic oxidation provides useful information for designing and optimizing electrochemical treatment systems. The recommendation to seek the most efficient conditions that minimize total costs while achieving high Diuron degradation is important for real-world applications. However, before publication, I have a few suggestions for improving the clarity and impact of your paper:
1. The introduction could be more comprehensive. There is a need for an overview of other technologies, such as photocatalysts and photoelectrolysis, as well as their environmental impact. Also, a brief discussion of the current treatment methods for Diuron and their limitations would be helpful. You may refer to the recent papers as follows, 1. Engineering low-coordination single-atom cobalt on graphitic carbon nitride catalyst for hydrogen evolution; 2.Restructuring highly electron-deficient metal-metal oxides for boosting stability in acidic oxygen evolution reaction; 3. Identifying the activity origin of a cobalt single‐atom catalyst for hydrogen evolution using supervised learning; 4. Unveiling Hierarchical Dendritic Co3O4–SnO2 Heterostructure for Efficient Water Purification; 5Promising energy-storage applications by flotation of graphite ores: A review 6. Identification of Dynamic Active Sites Among Cu Species Derived from MOFs@ CuPc for Electrocatalytic Nitrate Reduction Reaction to Ammonia 7. Construction of hollow binary oxide heterostructures by Ostwald ripening for superior photoelectrochemical removal of reactive brilliant blue KNR dye.
2. The discussion section could be expanded. There is a need to provide a more in-depth interpretation of the results and their implications. For instance, the author may add a comparison with recent literature.
3. Additionally, the limitations of your study and the potential for future research could be discussed. For instance, how to balance between electricity consumption and degradation performance.
Round 2
Reviewer 2 Report
Manuscript is sufficiently improved and can now be accepted for publication.
Manuscript is sufficiently improved and can now be accepted for publication.
Reviewer 4 Report
it well solved my concerns. it can be published as the current version.